# Novel DnaJ Protein Facilitates Thermotolerance of Transgenic Tomatoes

**DOI:** 10.3390/ijms20020367

**Published:** 2019-01-16

**Authors:** Guodong Wang, Guohua Cai, Na Xu, Litao Zhang, Xiuling Sun, Jing Guan, Qingwei Meng

**Affiliations:** 1College of Biological Science, Jining Medical University, Ri’zhao 276800, Shandong, China; xuna828@163.com (N.X.); zhanglt@mail.jnmc.cn (L.Z.); hatchslack@163.com (X.S.); guanjing1616@sina.com (J.G.); 2College of Life Science, Nanjing University, Nan’jing 210046, Jiangshu, China; caiguohua11@126.com; 3College of Life Science, State Key Laboratory of Crop Biology, Shandong Agricultural University, Tai’an 271018, Shandong, China

**Keywords:** DnaJ, heat stress, HSP70, SlDnaJ20, tomato

## Abstract

DnaJ proteins, which are molecular chaperones that are widely present in plants, can respond to various environmental stresses. At present, the function of DnaJ proteins was studied in many plant species, but only a few studies were conducted in tomato. Here, we examined the functions of a novel tomato (*Solanum lycopersicum*) DnaJ protein (SlDnaJ20) in heat tolerance using sense and antisense transgenic tomatoes. Transient conversion assays of *Arabidopsis* protoplasts showed that SlDnaJ20 was targeted to chloroplasts. Expression analysis showed that *SlDnaJ20* expression was induced by chilling, NaCl, polyethylene glycol, and H_2_O_2_, especially via heat stress. Under heat stress, sense plants showed higher fresh weights, chlorophyll content, fluorescence (Fv/Fm), and D1 protein levels, and a lower accumulation of reactive oxygen species (ROS) than antisense plants. These results suggest that *SlDnaJ20* overexpression can reduce the photoinhibition of photosystem II (PSII) by relieving ROS accumulation. Moreover, higher expression levels of *HsfA1* and *HsfB1* were observed under heat stress in sense plants, indicating that *SlDnaJ20* overexpression contributes to *HSF* expression. The yeast two-hybrid system proved that SlDnaJ20 can interact with the chloroplast heat-shock protein 70. Our results indicate that *SlDnaJ20* overexpression enhances the thermotolerance of transgenic tomatoes, whereas suppression of *SlDnaJ20* increases the heat sensitivity of transgenic tomatoes.

## 1. Introduction

Global agricultural production is affected by various environmental factors. With the gradual warming of global temperature, heat stress becomes more and more serious. The Intergovernmental Panel on Climate Change (2012) predicted that the global average temperature would rise by 2–5 °C at the end of the century. By then, many plants will not grow normally as a consequence of heat stress; therefore, cultivating new varieties resistant to heat stress and further studying the molecular mechanisms of plant response to heat stress are necessary. Heat-shock proteins (HSPs) are a kind of protein commonly found in plants in response to heat stress. Heat-resistant plants often enhance heat tolerance by overexpressing HSPs [1,2]. These proteins can generally be used in a number of processes to protect cells from heat stress, such as protein folding, protein export, DNA replication, and stress response [3,4].

DnaJ proteins, also known as heat-shock protein 40 (HSP40), perform the function of molecular chaperones independently or as the co-chaperone of HSP70 [5,6]. DnaJ protein generally contains four conserved domains, namely a J domain at the N-terminal, a glycine- and phenylalanine-rich region (G/F domain), a zinc finger(CxxCxGxG)4 domain, and a C-terminal domain [7,8]. Among them, the spatial structure of the J domain is composed of four helices, in which the second helix and the third helix are reverse parallel spatial structures, and the His/Pro/Asp (HPD) tripeptide with extremely conservative structure is located between the second and third helix. The G/F domain is a flexible linear structure, which may affect the function specificity of the J-protein. The most typical feature of zinc finger domain is that there are four CxxCxGxG repeat modules, which can interact with zinc ions and participate in the interaction between the J-protein and the target peptide. The most unconserved domain of the four domains is that at the C-terminal region, but this region has a similar three-dimensional structure consisting mainly of two β folds and a shorter α helix [9,10]. According to their conserved domains, DnaJ proteins are mainly divided into three groups: type I J-proteins contain all four domains (the J, G/F, zinc finger, and C-terminal domain), type II J-proteins do not contain a zinc-finger domain, and type III J-proteins only have a J domain [11,12]. Previous studies showed that J-proteins are involved in heat stress response in plants. Overexpression of both AtDjA2 and AtDjA3 enhances the heat resistance of *Arabidopsis* seedlings [13]. Thermosensitive male-sterile 1, as a J-protein, is important for the regular growth of *Arabidopsis* pollen tubes under heat stress [14]. AtDjB1 plays an important role in preventing cells from heat-induced oxidative damage in *A. thaliana* [15]. LeCDJ1 contributes to improving the heat tolerance of transgenic tomatoes [16]. *Solanum lycopersicum* chloroplast-targeted DnaJ protein (SlCDJ2) facilitates thermotolerance by maintaining Rubisco activity in transgenic tomato [17].

Heat stress often causes excessive accumulation of reactive oxygen species (ROS) in plants. Chloroplast is one of the important parts where ROS is produced in plants. However, excessive ROS can significantly suppress the de novo synthesis of D1 protein and, thus, affect the normal process of photosynthesis [18]. The enzymatic reaction and non-enzymatic reaction systems are the two main mechanisms that enable plants to remove excessive ROS. The non-enzymatic reaction system mainly refers to some non-enzymatic antioxidants, such as flavonoids, tocopherol, ascorbic acid, glutathione, carotene, mannitol, and so on. These substances can not only react with ROS and reduce them, but also act as a substrate of enzymes in the ROS clearance process. The enzymatic reaction system mainly includes three antioxidant enzymes, namely superoxide dismutase (SOD), ascorbate peroxidase (APX), and catalase (CAT). SOD, as the first line of defense in the plant antioxidant system, removes the superoxide anion in cells. SOD is divided into three types according to the different metal ions bound by its auxiliary base in plants: Cu/Zn-SOD, Mn-SOD, and Fe-SOD. Chloroplast mainly contains Cu/Zn-SOD and Fe-SOD. APX removes H_2_O_2_ and is divided into four types according to its location in plant cells: cytosolic APX (cAPX), microbody membrane-bound APX (mAPX), stromal APX (sAPX), and thylakoid membrane-bound APX (tAPX), where sAPX and tAPX exist in chloroplasts. CAT is mainly found in plant peroxidases, which also remove excess H_2_O_2_. Previous studies showed that the DnaJ protein plays an important role in protecting antioxidant enzyme activity and removing excess ROS [16,19].

The expression of many genes is changed in plants under heat stress. Heat stress transcriptional factors (HSFs) are some of the most important genes involved in response to heat stress, and they can recognize and bind specifically to the conserved motif of heat-shock element in the HSP gene promoter subregion to regulate the expression of the HSP gene. Plant HSF genes were first cloned in tomato [20]. At least 16 kinds of HSFs were found in tomatoes according to previous studies [21,22,23]. Among them, HsfA1, HsfA2, and HsfB1 are the three most representative and well-studied HSFs [20]. HsfA1a is the main regulatory factor in the expression of heat-induced genes and in the synthesis of HsfA2 and HsfB1 in tomatoes, which is essential in tomato heat resistance [24,25]. Hahn et al. demonstrated the direct interaction between Hsp70/Hsp90 and HSFs through yeast hybridization and immune co-precipitation experiments [26]. However, the role between DnaJ and HSFs in plant thermotolerance needs to be elucidated.

Therefore, we cloned and characterized a novel J-protein from tomato (SlDnaJ20), which is located in the chloroplast. SlDnaJ20 belongs to the simplest type III J-proteins characterized specifically by the typical J-domain. *SlDnaJ20* transcription was proven to be affected by heat stress. Moreover, *SlDnaJ20* overexpression in tomato reduced heat stress-induced photosystem damage, whereas *SlDnaJ20* suppression increased heat sensitivity.

## 2. Results

### 2.1. Identification and Bioinformatics Analysis of SlDnaJ20

*SlDnaJ20* was isolated and cloned from tomato leaves. The full length of *SlDnaJ20* complementary DNA (cDNA) is 1019 bp and contains a 591-bp open reading frame, which encodes a protein with 196 amino acids, with a calculated molecular weight of ∼22.7 kDa and a isoelectric point (pI) of 9.18 (https://web.expasy.org/cgi-bin/compute_pi/pi_tool). *SlDnaJ20* is located on tomato chromosome 5 according to the Sol Genomics Network (the number: SGN-U570442). The homologous comparison with reported LeCDJ1 and SlCDJ2 verified that SlDnaJ20 has typical HPD (His/Pro/Asp) motifs and belongs to the J-protein family (Figure 1A). However, according to the analysis from the phylogenetic tree of reported plant J-proteins and the characteristics of the SlDnaJ20 amino-acid sequence (Appendix A), SlDnaJ20 belongs to the simplest type III J-protein family (Figure 1B). SlDnaJ20 amino-acid sequence identities with LeCDJ1 and SlCDJ2 are 11.55% and 16.25%, respectively, indicating that the physiological function of SlDnaJ20 may be different from that of LeCDJ1 and SlCDJ2.

### 2.2. Subcellular Localization of SlDnaJ20

The databases TargetP 1.1 (http://www.cbs.dtu.dk/services/TargetP/) and ChloroP 1.1 (http://www.cbs.dtu.dk/services/ChloroP/) predicted that SlDnaJ20 may be a chloroplast protein (Appendix A). To confirm this prediction, transient conversion assays were conducted in vivo in *Arabidopsis* protoplasts extracted from leaf tissue with expressing 35S::enhanced GFP (*EGFP*) and 35S::*SlDnaJ20-EGFP* fusion protein (Figure 2A). As a contrast, not surprisingly, the green fluorescence of 35S::EGFP protein was dispersed in all parts of the protoplasts except the vacuoles (Figure 2B, upper panels). However, when the protoplasts were transfected with 35S::*SlDnaJ20-EGFP* fusion protein, green fluorescence signal was distinctly co-localized with the auto-fluorescent signal of chlorophyll in the chloroplasts (Figure 2B, lower panels). These results indicated that SlDnaJ20 is a chloroplast protein.

### 2.3. Expression Analysis of SlDnaJ20 in Tomato

*SlDnaJ20* was obtained from a gene tomato library whose gene expression patterns could be affected by heat stress. Therefore, the expression of SlDnaJ20 under heat stress was studied firstly via qPCR and Western blot methods. *SlDnaJ20* transcription levels were analyzed using qPCR at different temperatures (30 °C, 35 °C, 38 °C, 42 °C, and 45 °C) after 6 h of heat treatment. The expression of *SlDnaJ20* was the highest after the 42 °C treatment (Figure 3A). Therefore, the expression levels of *SlDnaJ20* during the study were determined with 42 °C for 24 h (Figure 3D). The transcription level reached its maximum at 6 h, which then decreased but recovered slowly to the original level after recovery at 25 °C for 6 h. Western blot results suggested that the change in protein signals of SlDnaJ20 was similar to that in the transcription levels (Figure 3B,E). Similarly, quantitative image analysis of SlDnaJ20 protein contents (Figure 3B,E) showed a similar profile (Figure 3C,F).

The expression level of *SlDnaJ20* was measured via qPCR analysis under chilling (4 °C), salt (400 mM NaCl), drought (25% polyethylene glycol (PEG-6000)), and oxidative (20 mM H_2_O_2_) stresses at different time points (Figure 4). The expression of *SlDnaJ20* was induced to different degrees under the abovementioned stress environments. The expression of *SlDnaJ20* induced via chilling stress first increased and then decreased, with the highest expression level at 12 h (Figure 4A). Salt stress induced the expression of *SlDnaJ20* similar to chilling stress, but the highest transcription level was observed at 9 h (Figure 4B). For osmotic stress, the expression of *SlDnaJ20* was induced gradually (Figure 4C). Moreover, the expression of *SlDnaJ20* induced via oxidative stress first decreased and then increased, with the highest expression level at 24 h (Figure 4D). These results indicated that *SlDnaJ20* is a multiple stress response gene.

The expression profiles of *SlDnaJ20* in different organs were analyzed via qPCR. Figure 4F shows that *SlDnaJ20* was constitutively expressed in various organs collected and preferentially in the leaves. Thus, these results suggested that *SlDnaJ20* might play its role mainly in chlorophyllous tissues.

### 2.4. Identification of Transgenic Plants

A total of 28 individual phosphinothricin-resistant transgenic tomato lines (T_2_) were collected from tissue culture, including 16 sense lines and 12 antisense lines. Six sense (S1, S2, S4, S6, S8, and S9) and antisense (A1, A3, A4, A5, A7, and A8) T_2_ lines were selected for qPCR to detect the expression level of *SlDnaJ20* in transgenic lines. Compared with wild-type (WT) plants, the *SlDnaJ20* transcription level in the tested sense pants increased by 12.9-, 65.0-, 44.9-, 32.2-, 25.7-, and 10.6-fold, whereas that in the antisense plants decreased by 0.37-, 0.20-, 0.47-, 0.27-, 0.30-, and 0.67-fold (Figure 5A). Among these transgenic lines, S2, S6, S9, A3, A5, and A8 were selected for Western blot analysis. The profile of SlDnaJ20 protein levels was similar to that of the transcript levels (Figure 5B,C). Therefore, S2, S6, S9, A3, A5, and A8 were selected for subsequent studies.

### 2.5. SlDnaJ20 Overexpression Enhanced Heat Stress Resistance

Given that *SlDnaJ20* expression was obviously induced by high temperature, growth performance of ten-day-old seedlings and six-week-old mature plants was observed under heat stress. Under natural conditions, both seedlings and mature plants grew normally, and there was no obvious difference in phenotype and physiological parameters among sense, WT, and antisense lines (Figure 6). After 42 °C treatment for two days, the growth of all seedlings was inhibited at varying degrees. The growth phenotype of seedlings showed a slight difference among sense, WT, and antisense lines (Figure 6A). However, compared with WT lines, leaf withering was more serious in antisense mature lines and less serious in sense mature lines after 42 °C treatment for 24 h (Figure 6B). Therefore, compared with WT lines, the sense lines had higher chlorophyll content and fresh weight, while the antisense lines had lower chlorophyll content and fresh weight (Figure 6C,D). Similarly, after heat treatment, compared with the WT, the net photosynthetic rate (*P*n) of antisense plants decreased significantly, while that of sense plants showed a slight decrease (Figure 6E). These results suggested that *SlDnaJ20* overexpression could enhance the thermotolerance of transgenic plants, while its suppression increased the thermal sensitivity of plants.

### 2.6. SlDnaJ20 Overexpression Alleviates ROS Accumulation by Maintaining High Levels of SOD and APX Activities

Heat stress usually causes the generation of ROS. H_2_O_2_ and O_2_^•−^, two main ROS species, were evaluated using 3′,3-diaminobenzidine (DAB) and nitroblue tetrazolium (NBT) staining, respectively. As Figure 7 shows, before treatment, H_2_O_2_ and O_2_^•−^ accumulations were relatively low and no obvious difference was observed between WT and transgenic lines. However, after heat stress for 12 h, the accumulation of brown polymerization product (DAB staining) increased, especially in WT and antisense lines, and the antisense plants had the deepest color (Figure 7A). Similar results were observed in O_2_^•−^ accumulation (Figure 7B). Quantitative determination of H_2_O_2_ and O_2_^•−^ revealed a similar result (Figure 7C,D). The above results showed that *SlDnaJ20* overexpression alleviates the accumulation of H_2_O_2_ and O_2_^•−^.

Under normal conditions, no obvious difference in SOD and APX activities was detected between WT and transgenic lines. After 42 °C treatment for 12 h, SOD and APX activities decreased in different degrees. However, compared with WT lines, the decrease range in SOD and APX activities in the sense lines was smaller, whereas that in antisense lines was larger (Figure 8A,B). To investigate the reason for changes in enzyme activity, the expressions of *SlCuZnSOD*, *SlFeSODSl*, *SlAPX1*, *SlAPX2*, and *SltAPX* were detected via qPCR. As shown in Figure 8C–G, under normal conditions, the expression levels of *SlCuZnSOD*, *SlFeSODSl*, *SlAPX1*, *SlAPX2*, and *SltAPX* showed no obvious difference among sense, WT, and antisense lines. After heat treatment for 12 h, the expression levels of *SlCuZnSOD*, *SlFeSODSl*, *SlAPX1*, *SlAPX2*, and *SltAPX* were drastically reduced, but there was also no obvious difference among sense, WT, and antisense lines. These results indicate that the low ROS accumulation in the sense plants was due to the high levels of SOD and APX activities via *SlDnaJ20* overexpression.

### 2.7. SlDnaJ20 Overexpression Alleviates Photoinhibition of Photosystem II (PSII) under Heat Stress

As a core protein subunit of photosystem II (PSII), D1 protein shows a rapid turnover and directly reflects the degree of photoinhibition. Western blot results suggested that D1 protein levels were not obviously different between WT and transgenic lines under natural growth conditions. After 24 h of heat stress, the protein contents of all the lines decreased; however, the decrease range was smaller in sense lines and larger in antisense lines compared with WT lines (Figure 9A,B). Fluorescence (Fv/Fm) was used to evaluate PSII photoinhibition. With the extension of heat processing time, the decrease in amplitude of Fv/Fm in antisense lines was significantly greater than that of WT, while the decrease in amplitude of Fv/Fm was the minimum in sense lines. In the recovery stage, the recovery rate of Fv/Fm in sense lines was significantly faster than that of WT lines, while the recovery rate of Fv/Fm was the slowest in antisense lines (Figure 9C). These results indicated that *SlDnaJ20* overexpression can alleviate PSII photoinhibition under heat stress.

### 2.8. SlDnaJ20 Overexpression Promotes Expression of HSFs under Heat Stress

HSFs are possibly involved in heat stress response. Therefore, the transcription levels of *HsfA1*, *HsfA2*, and *HsfB1* were evaluated via qPCR. As shown in Figure 10, the expression levels of *HsfA1*, *HsfA2*, and *HsfB1* showed no obvious difference among sense, WT, and antisense lines under natural conditions. After 12 h of heat stress, compared with WT, sense lines showed higher expression levels of *HsfA1* and *HsfA2* in response to heat stress, whereas no obvious difference was observed in antisense lines, indicating that *SlDnaJ20* overexpression may further enhance resistance to heat stress by promoting the expression of HSFs.

### 2.9. Interaction between SlDnaJ20 and Chloroplast Hsp70 (cpHsp70)

DnaJ proteins are the co-chaperones of Hsp70. However, LeCDJ1 and SlCDJ2 also interact with cpHsp70 [16,17]. Therefore, we postulated that SlDnaJ20 may also interact with cpHsp70. The interaction between SlDnaJ20 and cpHsp70 (GenBank No. EU195057.1) was analyzed using a yeast two-hybrid assay. The interaction between pGADT7-T and pGBKT7-53 proteins acted as a positive control. All yeast transformants grew normally on selective medium lacking Leu and Trp (SM-LW). The interaction between SlDnaJ20 and pGADT7 or cpHsp70 and pGBKT7 empty vectors was the negative control, and all yeast cells did not grow on SM-LWHA. However, when SlDnaJ20 was co-transformed with cpHsp70, blue colonies grew normally on selective medium lacking Leu, Trp, His, and adenine (SM-LWHA) (Figure 11), indicating that SlDnaJ20 interacted with cpHsp70.

## 3. Discussion

DnaJ protein is a widespread molecular chaperone in plants, which can maintain the homeostasis of proteins in plants under stress. So far, the function of J-proteins was identified in many plant species [27,28,29,30]. Approximately 63 J-proteins are found in tomato, of which only a few were studied in terms of their biological functions. LeCDJ1, a chloroplast J-protein, was proven to play an important role in maintaining PSII function under chilling stress, and *LeCDJ1* overexpression can enhance the heat resistance of transgenic tomatoes [16,31]. *LeCDJ2* overexpression could enhance drought tolerance and resistance to *Pseudomonas solanacearum* in transgenic tobacco [19]. In our study, a novel J-protein gene (*SlDnaJ20*) was identified from tomato. The results of amino-acid sequencing and phylogenetic tree analysis showed that SlDnaJ20 belongs to the simplest type III J-protein family (Figure 1). Transient transformation in *Arabidopsis* protoplasts proved that SlDnaJ20 was a chloroplast protein (Figure 2). However, SlDnaJ20, LeCDJ1, and LeCDJ2 were not highly homologous, which indicates that the biological function of SlDnaJ20 may be different from that of the two reported J-proteins. Expression pattern analysis showed that *SlDnaJ20* could respond to various abiotic stresses, especially heat stress (Figure 3 and Figure 4). The present study proved that SlDnaJ20 contributes to enhancing the heat tolerance of transgenic tomatoes.

Global warming brought great challenges to the survival of many plants, which need to evolve highly complex mechanisms to cope with heat stress. Sustained heat stress can disrupt cellular homeostasis, leading to the retardation of plant growth and development, and even death [32]. Photosynthesis is long considered as one of the most heat-sensitive processes in plants [33,34]. Heat stress inhibits photosynthetic activity because it breaks the redox balance and affects electron transfer [35]. The extra electrons bond with oxygen to form ROS, which are potentially dangerous for plants under heat stress. The experimental data in this study indicate that SlDnaJ20 contributed to alleviating the accumulation of ROS in transgenic plants under heat stress. After heat stress, sense plants exhibited not only lowered ROS content, but also higher chlorophyll content and *P*n value compared with WT and antisense plants (Figure 6 and Figure 7). The low accumulation of ROS in sense plants may be due to their increased SOD and APX activities. Moreover, the difference in SOD and APX gene expression among sense, WT, and antisense lines was not obvious after heat treatment (Figure 8). Therefore, the relatively high APX and SOD activities in the sense lines were not necessarily dependent on their transcription levels, and SlDnaJ20 as chaperone proteins may play a role in folding, unfolding, or assembly of these proteins. Chloroplasts are a main site of reactive oxygen products in higher green plants during abiotic stress [36]. Surplus ROS could damage the sensitive site of PSII and inhibit the de novo synthesis of D1 protein by suppressing the peptide elongation process [37]. Western blot results of D1 protein indicated that its decrease was the most serious in antisense plants after heat stress (Figure 9A,B). In addition to the D1 protein, Fv/Fm was also used to evaluate PSII photoinhibition. The experimental data showed that the Fv/Fm value of the sense plants decreased significantly with the prolonged heat treatment time, but the decrease in amplitude was smaller than that of WT and antisense plants (Figure 9C). The above results strongly suggest that the overexpression of *SlDnaJ20* can alleviate the photoinhibition of PSII by relieving ROS accumulation.

HSFs are key transcription factors in heat stress response, and they can directly activate HSPs, sHSPs, and other heat response proteins under heat stress. Increased expression of these genes can enhance the heat tolerance of plants [38,39,40,41]. This study revealed that the expressions of *HsfA1* and *HsfA2* in response to heat stress is higher in sense plants than in WT and antisense plants (Figure 10). These results indicate that *SlDnaJ20* overexpression enhances the heat tolerance of transgenic plants, probably because it promotes the expression of HSFs under heat stress. Moreover, Hahn et al. (2011) reported that a direct interaction exists between HSP70 and HSFs. As the co-chaperones of Hsp70, DnaJ proteins play an important role in stimulating Hsp70 ATPase activity. In this study, the yeast two-hybrid system proved that SlDnaJ20 can interact with tomato cpHSP70 (Figure 11). These findings suggest that the SlDnaJ20/cpHsp70 machinery possibly plays an important role in response to heat stress.

In conclusion, we isolated and cloned the novel tomato DnaJ gene *SlDnaJ20*. We found that the overexpression of *SlDnaJ20* contributed to alleviating ROS accumulation under heat stress, thereby reducing the photoinhibition of PSII. In addition, the overexpression of *SlDnaJ20* promoted the expression of HSFs to enhance the heat tolerance of transgenic plants, whereas its suppression increased the heat sensitivity of transgenic plants.

## 4. Materials and Methods

### 4.1. Plant Materials, Growth Conditions, and Stress Treatments

Wild-type (WT, *Solanum lycopersicum* cv. L-402) and six T_2_ transgenic lines (three sense lines, S2, S6, and S9; three antisense lines, A3, A5, and A8) were grown in sterilized soil in the following conditions: 55–65% relative humidity, 25 °C, 16 h/8 h (light/dark), and 250 µmol⋅m^−2^⋅s^−1^ photon flux density (PFD). These plants were watered with Hoagland’s nutrient solution twice a week. When they grew to about six weeks and the sixth leaf was fully unfolded, the plants were transferred to the illuminated growth chamber (GXZ-500C) for 2–3 days before treatment.

In order to investigate the expression of *SlDnaJ20* under different stresses, six-week-old tomato WT plants were treated under different stresses. For 4 °C treatments, the plants were treated under 4 °C for 0, 3, 6, 9, 12, and 24 h under a 16-h (light)/8-h (dark) regime. For NaCl, polyethylene glycol (PEG), and H_2_O_2_ treatments, the roots of WT plants were dipped in Hoagland’s nutrient solution containing 400 mM NaCl, 25% PEG 6000 (*w*/*v*), and 20 mM H_2_O_2_ for 0, 3, 6, 9, 12, and 24 h, whereas the control plants were only watered with Hoagland’s nutrient solution. For heat treatments, WT plants were subjected to 30 °C, 35 °C, 38 °C, 42 °C, and 45 °C for 6 h, and then the plants were also treated under 42 °C for 0, 3, 6, 9, 12, and 24 h. Meanwhile, ten-day-old seedlings were treated at 42 °C for two days, and six-week-old WT and transgenic plants were subjected to 42 °C for 12 and 24 h in an illuminated growth chamber with approximately 250 µmol⋅m^−2^⋅s ^−1^ PFD for physiological parameter detection.

### 4.2. Isolating and Sequencing of SlDnaJ20

The coding sequence (CDS) of *SlDnaJ20* was amplified with cDNA from WT tomato leaves using specific primers (forward 5′–ATGTGTTGCAACTCCAATGG–3′ and reverse 5′–TTATGCATCATCATCCCTTT–3′) according to the messenger RNA (mRNA) sequence (GenBank No. XM_004239806) by polymerase chain reaction (PCR). Thereafter, the PCR amplification products were connected to pMD19-T vector (TaKaRa, Beijing, China) and sequenced. Protein multiple sequence alignments were performed using ClustalW (http://www.genome.jp/tools/clustalw/). The phylogenetic relationship of SlDnaJ20 protein was made using MAGE 5.1 software (https://mega.software.informer.com/5.1b).

### 4.3. Subcellular Localization of SlDnaJ20

The complete CDS of *SlDnaJ20* was amplified with specific primers (forward 5′–CTCGAGATGTGTTGCAACTCCAATGG–3′ and reverse 5′–GGTACCGTTGCATCATCATCCCTTTGCT–3′). Subsequently, the *SlDnaJ20* coding region was ligated into the reconstructed binary vector pEZS-NL digested with *Xho*I/*Kpn*I, which generated a *SlDnaJ20-EGFP* (enhanced green fluorescent protein) construct. The *EGFP* alone and *SlDnaJ20::EGFP* (recombinant plasmids) were transformed into *Arabidopsis* mesophyll protoplasts, and then laser confocal microscopy (LSM510 META; Zeiss, Oberkochen, Germany) was used to examine the fluorescent.

### 4.4. Tomato Genetic Transformation and Identification

We inserted the CDS of *SlDnaJ20* into the pCAMBIA3301 binary expression vector under the control of the CaMV 35S promoter. The CDS of *SlDnaJ20* was also inserted into the expression vector pCAMBIA3301 inversely. The recombinant plastids were transferred into *Agrobacterium tumefaciens* LBA4404 via freezing transformation method and confirmed using PCR and sequencing analyses. The genetic transformation of tomato plants was performed following the *A. tumefaciens*-mediated leaf disc method. T_0_ phosphinothricin-resistant transgenic tomato plants were generated by the *A. tumefaciens* mediated leaf disk method. After selection for phosphinothricin-resistant T_0_ transgenic plants, PCR-based genotyping was performed to further identify the sense and antisense T_0_ transgenic plants. T_2_ progeny homozygotes with 100% phosphinothricin resistance were used in the following experiments.

### 4.5. Real-Time Quantitative PCR (qPCR) Analysis

Total RNA was isolated from transgenic and WT tomato leaves using Trizol reagent (TIANGEN, Beijing, China). Quantitative PCR was performed following the method described by Zhang et al. [42]. *EF-1*α (GenBank No. LOC544055) acted as an actin control. The primers used for qPCR are listed in Appendix A.

### 4.6. Antibody Production, Protein Acquisition, and Protein Level Analysis

The CDS of *SlDnaJ20* was ligated into the pET-30a (+) vector digested with *BamH*I/*Sac*I. SlDnaJ20 antibody was produced as described previously [19]. Total protein was isolated from the leaves according to the method described previously [31]. Thylakoid membranes were isolated following the method described by Zhang et al. [43]. Western blot analysis was performed following the method described previously [17]. 

### 4.7. Measurement of Net Photosynthetic Rate (Pn) and Chlorophyll Fluorescence

*P*n was detected using a CIRAS-3 Portable Photosynthesis System (PP Systems, Amesbury, MA, USA) under ambient 380 µL·L^−1^ CO_2_ conditions, 80% relative humidity, 800 µmol·m^−2^·s^−1^ PFD and 25 °C leaf temperature.

The chlorophyll fluorescence was detected with an FMS-2 pulse-modulated fluorometer (Hansatech Instruments, Norfolk, UK). Before testing, plants needed to be dark-adapted for 30 minutes. The measurement was conducted as described in Jiang et al. [44]. The *F*v/*F*m of PSII was calculated as follows: *F*v/*F*m = (*F*m–*F*o)/*F*m.

### 4.8. Histochemical Staining and Measurements of H_2_O_2_ and O_2_^•−^

Hydrogen peroxide (H_2_O_2_) and superoxide radical (O_2_^•−^) were stained with 3′,3-diaminobenzidine (DAB) and nitroblue tetrazolium (NBT), respectively. The staining protocols were performed as described in Pan et al. [45]. The contents of H_2_O_2_ and O_2_^•−^ in WT and transgenic plant leaves were detected following previous research methods [31].

### 4.9. Measurements of Chlorophyll Content and Antioxidative Enzyme Activities

The chlorophyll content of the WT and transgenic plant leaves was detected following the method described in Kong et al. [31]. SOD and APX were measured in the leaves following the method described in Zong et al. [46].

### 4.10. Yeast Two-Hybrid Assays

The CDS of *SlDnaJ20* was ligated to pGADT7 (Clontech, Palo Alto, CA, USA) digested with *EcoR*I/*Sac*I. The CDS of *cpHsp70* (GenBank No. EU195057.1) was amplified with cDNA from tomato leaves and inserted to pGBKT7 (Clontech, Palo Alto, CA, USA) digested with *BamH*I/*Xho*II. Then, the pGADT7-*SlDnaJ20* recombinant plasmid was co-transformed with pGBKT7-*cpHsp70* into the yeast strain Y187 (Clontech, Palo Alto, CA, USA) via the lithium acetate transformation method (Clontech Yeast Two-Hybrid User Manual). Yeast cells were coated onto a selective medium lacking Leu and Trp (SM-LW). Putative transformants were transferred to a selective medium lacking Leu, Trp, His, and adenine, but with the addition of X-α-Gal and aureobasidin (SM-LWHA/X/A). The interaction between pGADT7-T and pGBKT7-53 proteins was used as a positive control. The result was based on three independent biological repeats.

### 4.11. Statistical Analysis

SigmaPlot 12.5 (Systat Software, San Jose, CA, USA) and SPSS13.0 (Chicago, IL, USA) were used for statistical analyses. The mean values ± SD of at least three replicates are presented, and * *p* < 0.05 and ** *p* < 0.01 indicate significant differences compared with the control.

## Figures and Tables

**Figure 1 ijms-20-00367-f001:**
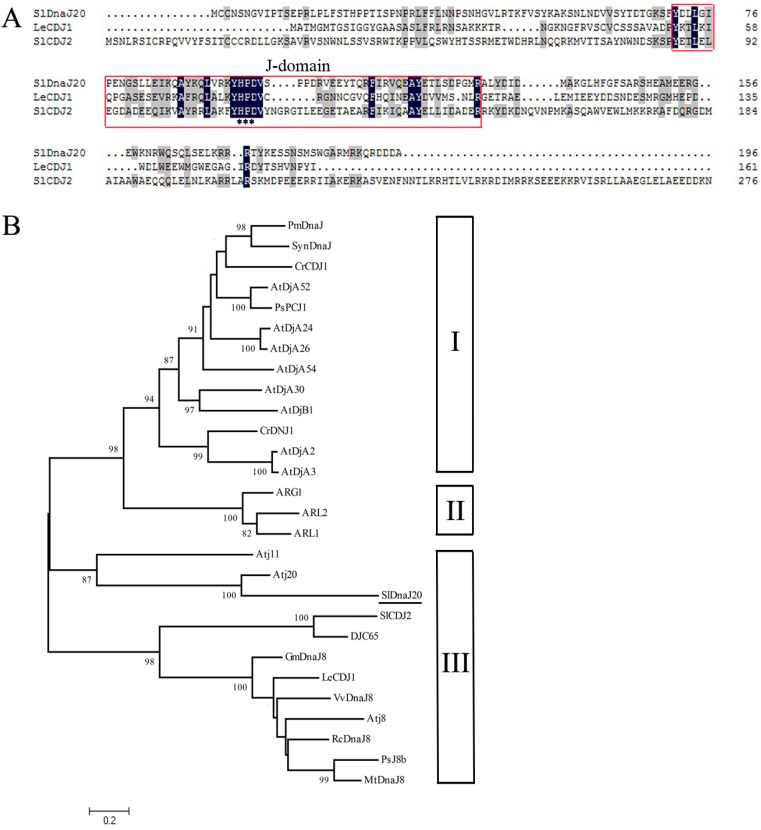
Multiple sequence alignment and phylogenetic tree analysis of *Solanum lycopersicum* DnaJ protein (SlDnaJ20). (**A**) Multiple sequence alignment between SlDnaJ20 and other J-proteins in tomato. Black and gray refer to identical and similar amino acids, respectively. The His/Pro/Asp (HPD) motifs are marked by stars. The protein sequence of SlDnaJ20 only contains a J-domain (red box). (**B**) Phylogenetic tree analysis of SlDnaJ20 with other J-proteins. Roman numerals I–III on the right represent different types of J-proteins. SlDnaJ20 is underlined. The phylogenetic tree of various J-proteins generated with the ClustalW2 program using the Neighbor-Joining method in MEGA (version 5.1) (https://mega.software.informer.com/5.1b). Bootstrap analyses were computed with 1000 replicates, the percentage values larger than 80 are shown in the branches. The gene names and GenBank accession numbers are as follows: PmDnaJ, YP_001013845; SynDnaJ, ZP_01084411; CrCDJ1, AAU06580; AtDjA52, AAD55483; PsPCJ1, CAA96305; AtDjA24, NP_568076; AtDjA26, NP_565533; AtDjA54, NP_188410; atDjA30, BAB11067; AtDjB1, At1g28210; CrDNJ1, EDP04706; AtDjA2, At5g22060; AtDjA3, At3g44110; ARG1, AEE34786; ARL2, At1g59980; ARL1, At1g24120; AtJ11, At4 g36040; AtJ20, At4g13830; SlDnaJ20, XM_004239806; SlCDJ2, AK323942; DJC65, At1g77930; GmDnaJ8, ACU18989; LeCDJ1, AK323422; VvDnaJ8, XP_002263153; AtJ8, At1g80920; RcDnaJ8, EEF49240; PsJ8b, ADL32216; and MtDnaJ8, ACJ83936.

**Figure 2 ijms-20-00367-f002:**
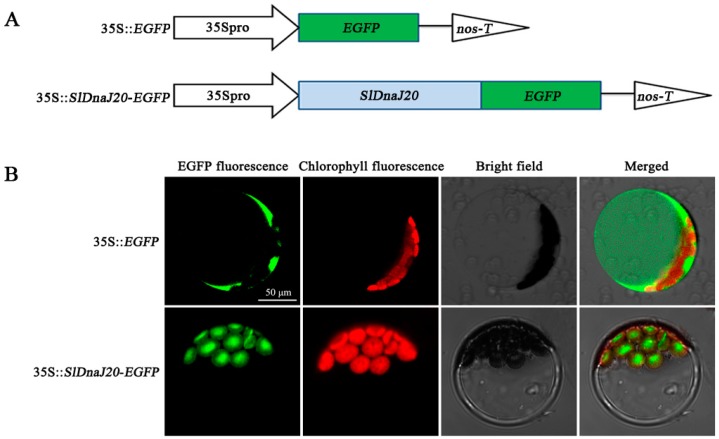
Subcellular localization of SlDnaJ20. (**A**) Pattern of enhanced GFP (EGFP) fusion protein structure. 35Spro, CaMV35S promoter. nos-T, nopaline synthase terminator. (**B**) 35S::*EGFP* (upper panels) and 35S::*SlDnaJ20-EGFP* (lower panels) were transiently expressed in *Arabidopsis* protoplast. Protoplasts were examined under laser confocal microscopy.

**Figure 3 ijms-20-00367-f003:**
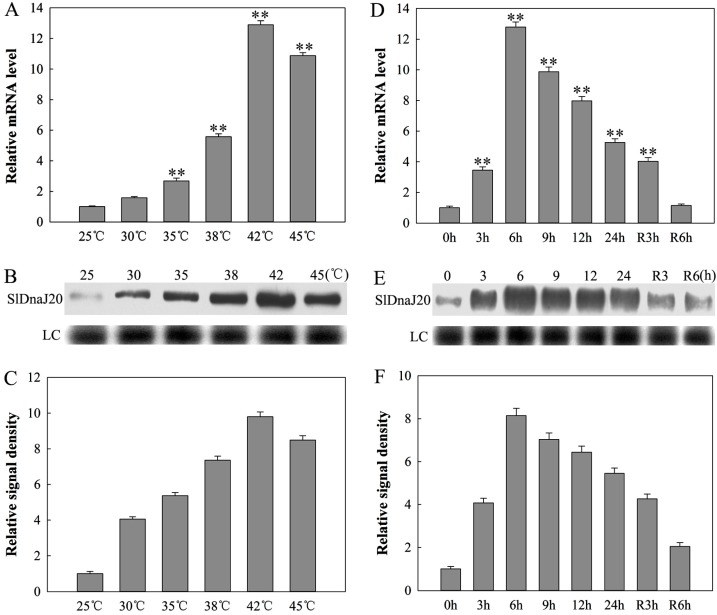
Expression of *SlDnaJ20* was induced by heat stress. (**A**) Quantitative PCR analysis of *SlDnaJ20* expression under different temperatures for 6 h. (**B**) Western blot analysis of SlDnaJ20 protein levels under different temperatures for 6 h. LC, loading control (part of a Coomassie-stained total protein SDS polyacrylamide gel). (**C**) Image quantification of protein content in (**B**). (**D**) Quantitative PCR analysis of *SlDnaJ20* expression in leaves subjected to 42 °C for 0, 3, 6, 9, 12, and 24 h, and recovered for 3 and 6 h. (**E**) Western blot analysis of SlDnaJ20 protein levels in leaves subjected to 42 °C for 0, 3, 6, 9, 12, and 24 h, and recovered for 3 and 6 h. LC, loading control (part of a Coomassie-stained total protein SDS polyacrylamide gel). **(F**) Image quantification of the protein content in (**E**). Columns (**A**) and (**D**) represent the mean ± SD of three replicates. Statistically significant differences with respect to the control are indicated as: ** *p* < 0.01.

**Figure 4 ijms-20-00367-f004:**
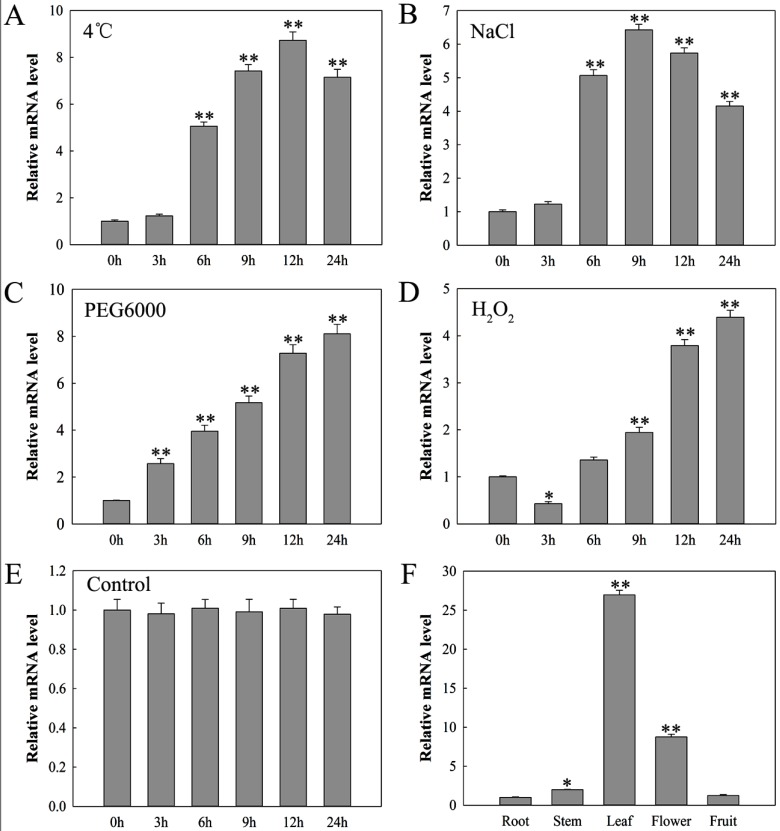
Quantitative PCR analysis of the expression profiles of *SlDnaJ20* in tomato: (**A**) 4 °C, (**B**) 250 mM NaCl, (**C**) 20% polyethylene glycol (PEG), (**D**) 10 mM H_2_O_2_, (**E**) control, (**F**) expression of *SlDnaJ20* in different tomato tissues. The mean values ± SD of at least three replicates are presented, and * *p* < 0.05 and ** *p* < 0.01 indicate significant differences compared with the control.

**Figure 5 ijms-20-00367-f005:**
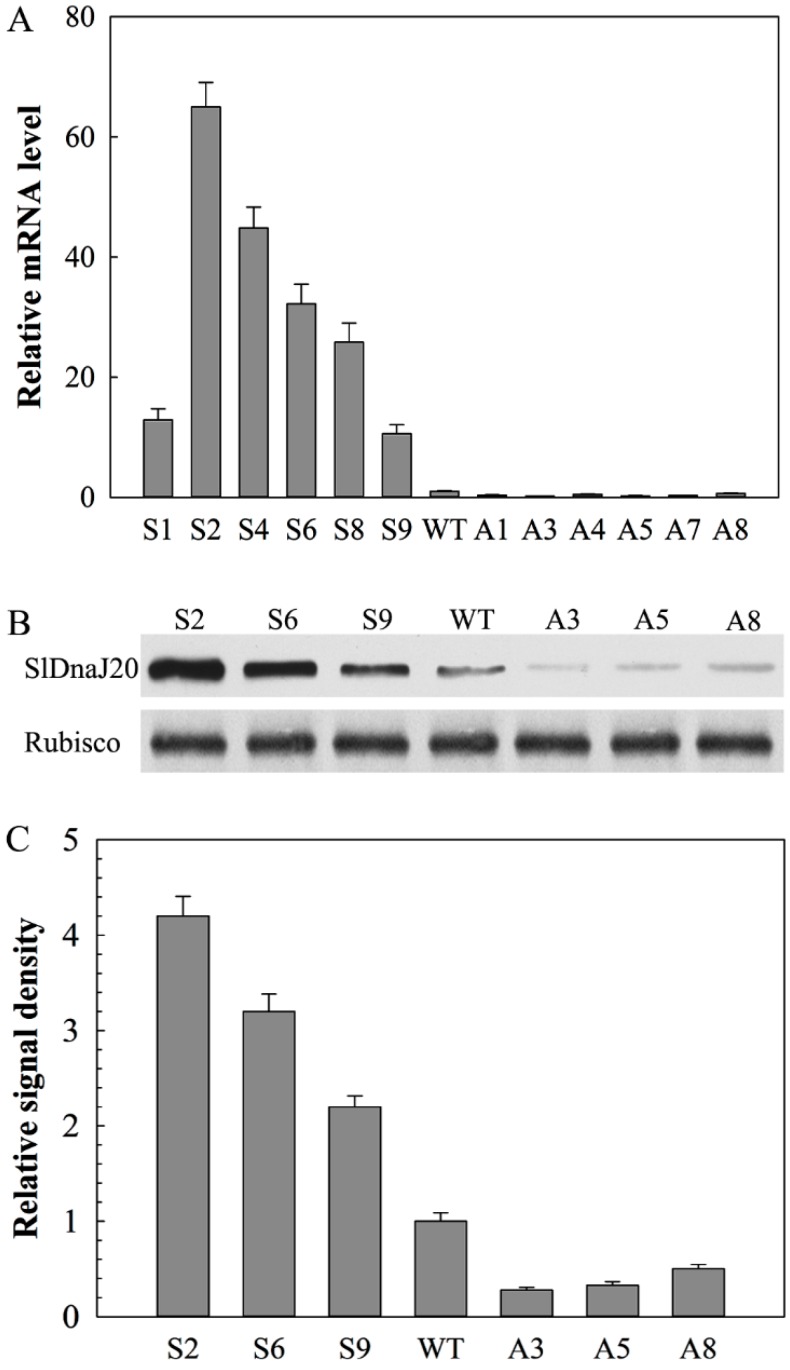
Identification of transgenic plants via qPCR and Western blot analysis. (**A**) *SlDnaJ20* transcript levels in wild-type (WT), sense, and antisense plants. (**B**) SlDnaJ20 protein levels in WT, sense, and antisense plants. The loading control is the large subunit of Rubisco. (**C**) Image quantification of protein contents in (**B**).

**Figure 6 ijms-20-00367-f006:**
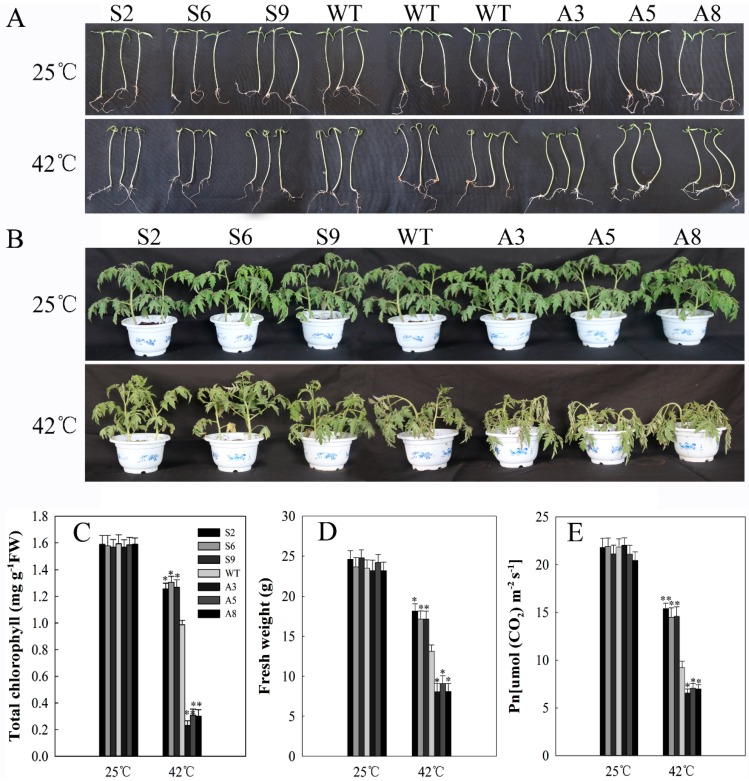
Heat tolerance of ten-day-old seedlings and six-week-old plants. (**A**) Phenotype of ten-day-old seedlings under 25 °C and 42 °C for two days; (**B**) phenotype of six-week-old plants under 25 °C and 42 °C for 24 h; (**C**) total chlorophyll content in grown plants; (**D**) fresh weight of grown plants. (**E**) net photosynthetic rate (*P*n) in the grown plants. The mean values ± SD of at least three replicates are presented, and * *p* < 0.05 and ** *p* < 0.01 indicate significant differences compared with the control.

**Figure 7 ijms-20-00367-f007:**
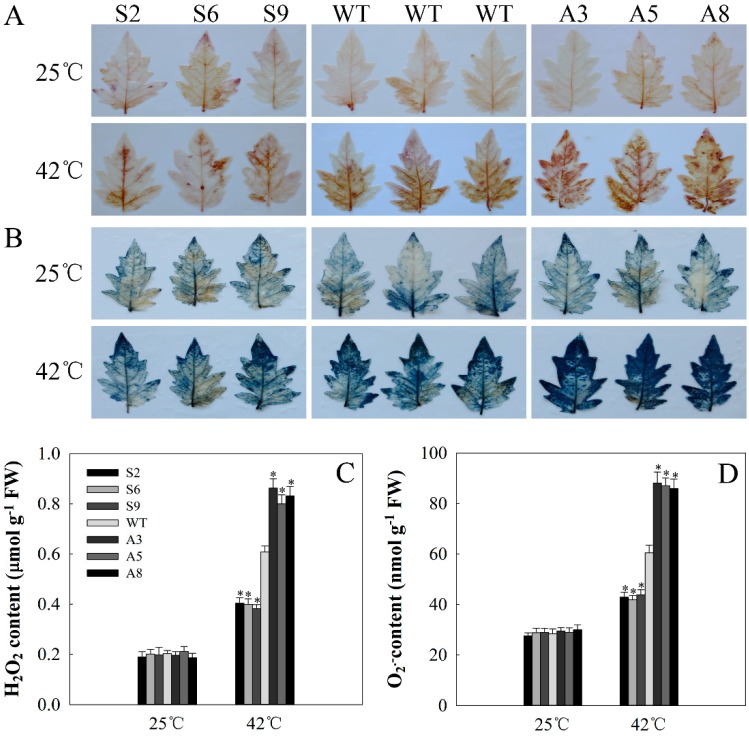
Reactive oxygen species (ROS) analysis in WT and transgenic lines: (**A**) 3′,3-diaminobenzidine (DAB) staining for H_2_O_2_; (**B**) nitroblue tetrazolium (NBT) staining for O_2_^•−^. The upper layer represents plants grown at 25 °C and the lower layer represents plants subjected to 42 °C for 12 h. (**C**) H_2_O_2_ content; (**D**) O_2_^•−^ content. The mean values ± SD of at least three replicates are presented, and * *p* < 0.05 indicate significant differences compared with the control.

**Figure 8 ijms-20-00367-f008:**
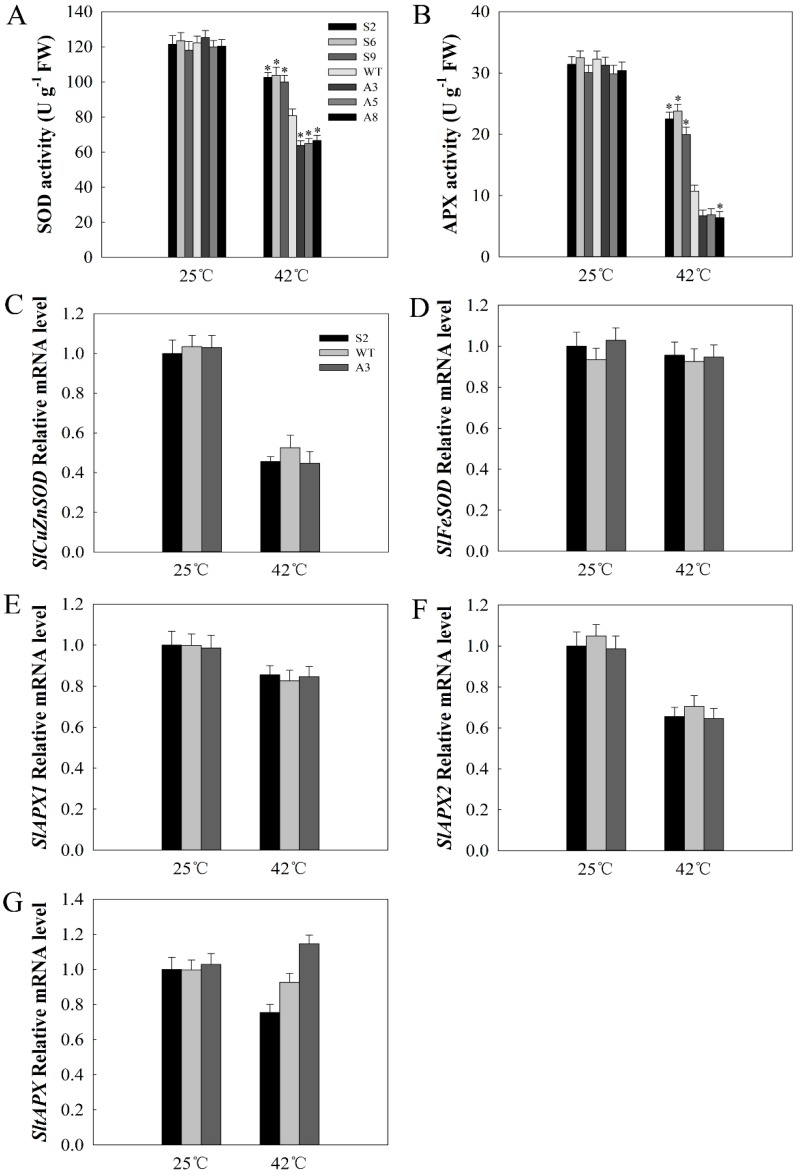
Analysis of antioxidant enzyme activity and related gene expression. (**A**) Superoxide dismutase (SOD) activity; (**B**) ascorbate peroxidase (APX) activity; (**C**) *SlCuZnSOD* expression; (**D**) *SlFeSOD* expression; (**E**) *SlAPX1* expression; (**F**) *SlAPX2* expression; (**G**) *SlTAPX* expression. The mean values ± SD of at least three replicates are presented, and * *p* < 0.05 indicate significant differences compared with the control.

**Figure 9 ijms-20-00367-f009:**
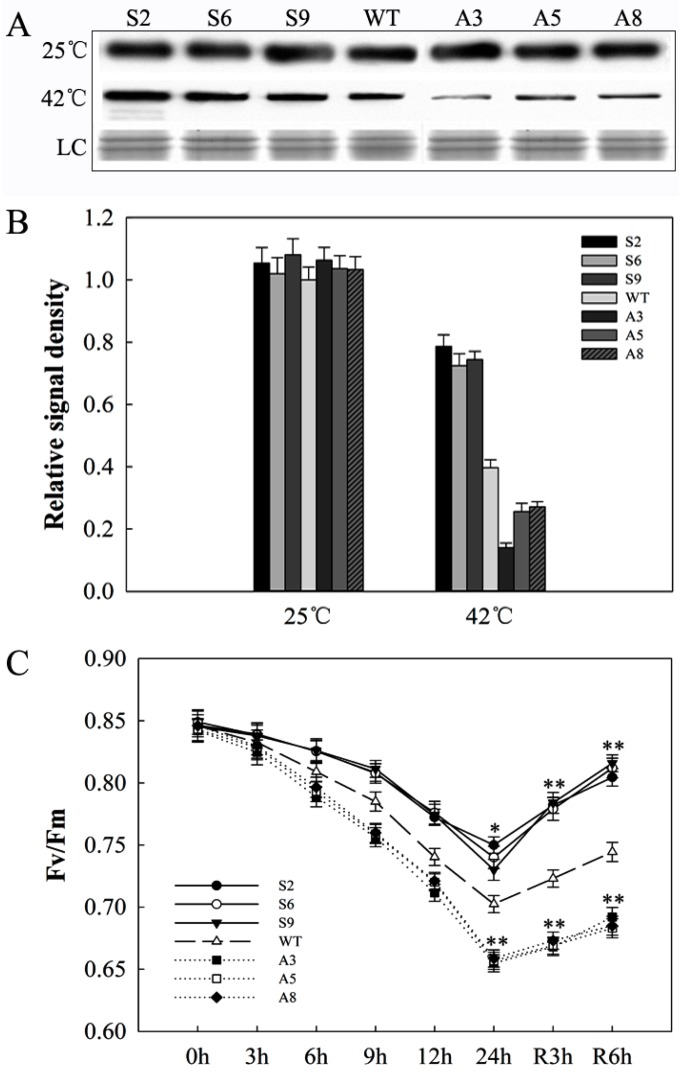
Changes in D1 protein levels and fluorescence (Fv/Fm) in WT and transgenic plants under heat stress. (**A**) D1 protein content. LC, loading control (part of a Coomassie-stained thylakoid membrane protein SDS polyacrylamide gel). (**B**) Image quantification of protein contents in (**B**). (**C**) Changes in Fv/Fm during heat stress and recovery. The mean values ± SD of at least three replicates are presented, and * *p* < 0.05 and ** *p* < 0.01 indicate significant differences compared with the control.

**Figure 10 ijms-20-00367-f010:**
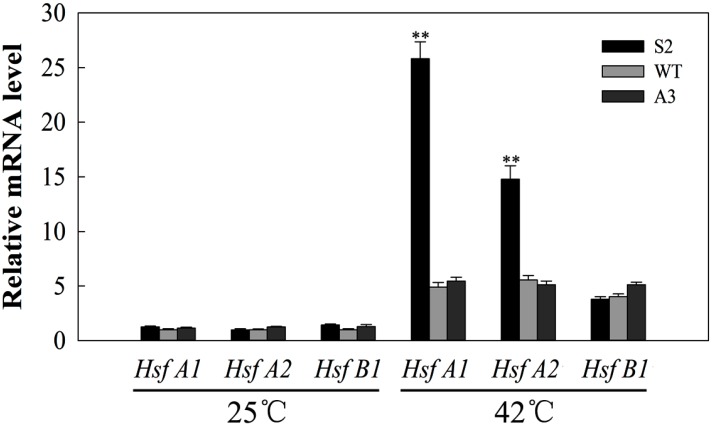
Quantitative PCR analysis of *HsfA1*, *HsfA2*, and *HsfB1* expression before and after heat stress. Six-week-old WT and transgenic plants were used for treatment under 42 °C for 12 h. The mean values ± SD of at least three replicates are presented, and ** *p* < 0.01 indicate significant differences compared with the control.

**Figure 11 ijms-20-00367-f011:**
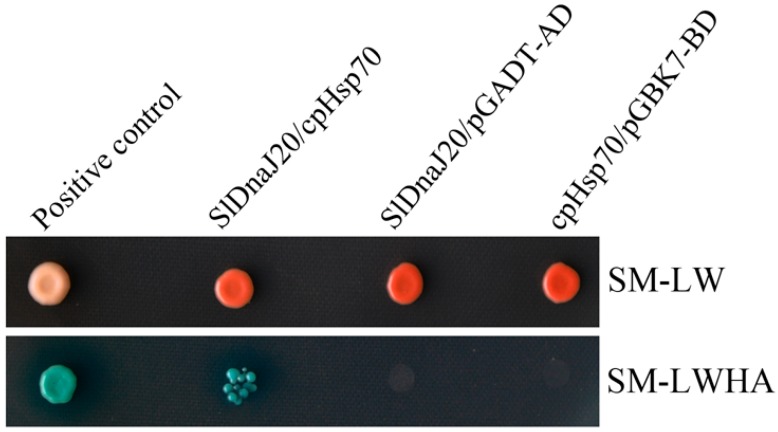
The interaction between SlDnaJ20 and chloroplast heat-shock protein 70 (cpHsp70) was analyzed using a yeast two-hybrid system. pGADT7-T and pGBKT7-53 proteins were co-transformed as a positive control. SlDnaJ20/pGADT-AD and cpHsp70/pGBK7-BD were used as negative controls. SM-LW indicates selective medium lacking Leu and Trp, and SM-LWHA indicates medium lacking Leu, Trp, His, and adenine with X-α-Gal.

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
