# Peer review of "Novel DnaJ Protein Facilitates Thermotolerance of Transgenic Tomatoes"

_ijms, 2019, doi:10.3390/ijms20020367_

Round 1

Reviewer 1 Report

The manuscript ID: ijms-419966 deals with the functions of a tomato DnaJ protein (SlDnaJ20) in heat tolerance, this is an interesting work. Different physiological and biochemical parameters are used to evaluate plant tolerance of sense and antisense transgenic tomatoes to the adverse temperature conditions. The paper is well written, the results are clearly presented and are easy for understanding, the conclusions are supported by the results. The authors identified the important role of SlDnaJ20 in heat tolerance of plants. The manuscript can be accepted for publication to IJMS.

Author Response

Response to Reviewer 1 Comments

Point 1: The manuscript ID: ijms-419966 deals with the functions of a tomato DnaJ protein (SlDnaJ20) in heat tolerance, this is an interesting work. Different physiological and biochemical parameters are used to evaluate plant tolerance of sense and antisense transgenic tomatoes to the adverse temperature conditions. The paper is well written, the results are clearly presented and are easy for understanding, the conclusions are supported by the results. The authors identified the important role of SlDnaJ20 in heat tolerance of plants. The manuscript can be accepted for publication to IJMS.

Response 1: Thank you very much for your recognition and praise

Reviewer 2 Report

Manuscript ID: ijms-419966

Title: Novel DnaJ protein facilitates thermotolerance of transgenic tomatoes

Authors: Guodong Wang, Guohua Cai, Na Xu, Litao Zhang, Xiuling Sun, Jing Guan, Qingwei Meng

Summary

            DnaJ proteins represent a family of molecular chaperones necessary for maintenance of cellular proteostasis under normal as well as stress conditions. The authors have presented a study that demonstrates a connection between Solanum lycopersicum DnaJ20 (SlDnaJ20) and thermotolerance. Solanum lycopersicum is a novel strain of tomato plant. Based on the data presented here, the authors propose a role for DnaJ20 in protecting tomato crops against heat stress presented by climate change. Analysis of mRNA and protein expression levels suggest that DnaJ20 overexpression is induced by exposure to cold temperatures, NaCl, polyethylene glycol, H2O2, and heat stress. Therefore, SlDnaJ20 most likely participates in the protecting the cell against varied stressors.

Major Comments

·        Overall, the authors have demonstrated clearly that a correlation exists between SlDnaJ20 overexpression and stress tolerance (specifically thermotolerance). However, the authors do not clearly demonstrate causation. One possible alternative explanation of the data is that SlDnaJ20, either alone or in complex with cpHsp70, simply reinforces the folding of many proteins that collectively contribute to thermotolerance. SlDnaJ20 likely participates in a network of chaperones and proteases that collectively function to insure that all proteins critical to cellular function are functioning. This is contrast to the viewpoint present by the authors wherein they imply that SlDnaJ20 specifically targets SOD and APX proteins. If the authors wish to assert this, then they need to demonstrate that SOD and APX are both substrates of SlDnaJ20.

o   To determine whether SlDnaJ20 is directly responsible for thermotolerance, the authors should consider generating a knockout or knockdown strain from sense strain. The comparison of phenotype with and without SlDnaJ20 will provide insight critical to the determination of correlation versus causation.

·        The authors have decided to pursue studies focused on the characterization of sense, wild-type, and antisense strains of Solanum lycopersicum. However, no data has been presented to clearly indicate whether the SlDnaJ20 primary sequence is identical in all strains examined. For example, comparison of strains S1 and S2 in Figure 5A suggest greater than 3-fold difference in SlDnaJ20 transcription. Reference to Figures 7C and 7D also suggest that sense strains with very different transcription levels yield statistically identical H2O2 and superoxide content. Thus, greater mRNA transcription does not automatically yield increased ROS buffering. As such, the authors must demonstrate that the primary sequence for SlDnaJ20 homologues from each tomato strain are identical to insure that their functional output are approximately equivalent. If mutations are found, some effort needs to be made to assess differences in enzyme function for wild-type versus mutant DnaJ20.

Minor Comments

·         Introduction:

o   At some points, the introduction reads as an alphabet soup—Protein A interacts with Protein B. This pattern repeats such that it can difficult to follow the introduction and points are not clearly connected sometimes. As such, the authors should take care to format their introduction to better connect all points.

o   The introduction does not clearly describe DnaJ protein structure, though this is used to differentiate Types I-III DnaJ proteins. The authors must describe the J domain, zinc finger, and C-terminal with regards to overall structure and function. Moreover, what catalytic motifs are harbored within each domain? Clarification on these points will significantly enhance the reader’s experience.

o   Page 3, Lines 95-96: The authors state that differences in primary sequence identities can be used to predict differences in physiological function between SlDnaJ20, LeCDJ1 and SlCDJ2. This assumption may be correct, but should not be assumed to be universally true. This prediction should be advanced by a more detailed discussion of differences in the conservation of known catalytic motifs shared between these proteins.

o   The authors conclude that SlDnaJ20 belongs to the Type III J-Protein family based on their phylogenetic analysis.

§  The manuscript does not provide sufficient information for the reader to ascertain the validity of the phylogenetic tree. No description of the phylogenetic analysis, software, methods, etc. are provided in the Materials and Methods section. No statistical parameters or bootstrapping information are provided to assess tree quality. If the authors plan to present such phylogenetic data, they need to provide the necessary information for the reader to determine the likelihood that the SlDnaJ20 branch is correctly placed.

§  No primary sequence information is provided to indicate that SlDnaJ20 lacks a zinc finger and C-terminal domain. Please provide this data (at least as a supplemental document).

o   Page 1, line 35: “By then, many plants will not grow normally;’ ---- Should this read “will not grow normally as a consequence of heat stress” or “because of climate change.” ?

o   Page 1, Line 34: should read “2-5 °C”

o   Page 2, Lines 53-67: Authors might consider expanding this paragraph to address the deleterious effects that can occur when redox buffering systems are exhausted. An example can be found in heart failure where mitochondrial redox buffering systems are overwhelmed, thereby contributing to increased ROS formation and organelle damage.

o   Page 2, Lines 65-67: As written, this sentence is misleading in that it implies that DnaJ directly removes ROS. This sentence needs to be restructured to clarify that DnaJ function is necessary to insure proper function of a separate protein that catalyzes ROS removal. That is, DnaJ does not catalyze the removal of ROS directly.

o   Page 2, Lines 71-72: “The plant HSF…” This reads as though there is only one gene in plants.

o   Page 2, Line 91: Clarify what HPD motifs are. Also, need to elaborate on J-family member requirements.

o   Page 3, Figure 1: How many bootstrap replicates and what method of analysis used for the phylogenetic analysis?

o   Page 5, Figures 3B, 3E: What protein is used as the loading control?

o   Page 5, Figure 3: How many replicates are reflected by error bars (mention in figure legend)? A comparison of means (T-test) is needed to convincingly demonstrate that values are different.

o   Page 6, Figure 4C: Though explained in the text, this figure refers to PEG6000 simply as PEG. Please consistently refer to PEG6000 as such, not just PEG.

o   Page 10, Figure 8G: For SltAPX mRNA levels, there may be a difference between S2, WT, and A3 strains at elevated temperature. On the previous page, the authors state that there is no difference, but the figure seems to suggest otherwise. Please perform T-testing to determine if the means are truly different or not.

o   Page 11, Figure 9A: What is the protein used as a loading control?

o   Page 12, Figure 11: Please describe why pGADT-AD and pGBK7-BD were chosen for this assay? What are these proteins?

Author Response

Please see the below attachment.

Round 2

Reviewer 2 Report

With the newly incorporated changes, the manuscript is now suitable for publication in IJMS.